# Hot Deformation Behavior and Processing Maps of a New Ti-6Al-2Nb-2Zr-0.4B Titanium Alloy

**DOI:** 10.3390/ma14092456

**Published:** 2021-05-09

**Authors:** Zhijun Yang, Weixin Yu, Shaoting Lang, Junyi Wei, Guanglong Wang, Peng Ding

**Affiliations:** School of Mechanical and Electrical Engineering, Xinxiang University, Xinxiang 453003, China; yuweixin2012@163.com (W.Y.); shaotinglang@163.com (S.L.); ai_tongle@126.com (J.W.); wangglo@163.com (G.W.); dingpeng@xxu.edu.cn (P.D.)

**Keywords:** titanium alloy, thermal deformation, constructive equation, processing maps

## Abstract

The hot deformation behaviors of a new Ti-6Al-2Nb-2Zr-0.4B titanium alloy in the strain rate range 0.01–10.0 s^−1^ and temperature range 850–1060 °C were evaluated using hot compressing testing on a Gleeble-3800 simulator at 60% of deformation degree. The flow stress characteristics of the alloy were analyzed according to the true stress–strain curve. The constitutive equation was established to describe the change of deformation temperature and flow stress with strain rate. The thermal deformation activation energy Q was equal to 551.7 kJ/mol. The constitutive equation was $\dot{\varepsilon \;}{=e}^{54.41}{[\sin h\;(0.01\sigma )]}^{2.35}\exp(-551.7/RT).$ On the basis of the dynamic material model and the instability criterion, the processing maps were established at the strain of 0.5. The experimental results revealed that in the (α + β) region deformation, the power dissipation rate reached 53% in the range of 0.01–0.05 s^−1^ and temperature range of 920–980 °C, and the deformation mechanism was dynamic recovery. In the β region deformation, the power dissipation rate reached 48% in the range of 0.01–0.1 s^−1^ and temperature range of 1010–1040 °C, and the deformation mechanism involved dynamic recovery and dynamic recrystallization.

## 1. Introduction

Titanium alloys are important structural materials in ship and aerospace due to high-temperature resistance, the high specific strength, weldability, corrosion resistance, and other excellent characteristics [1,2,3,4]. In recent years, an urgent need to reduce the weight of structural materials in aerospace is observed. High strength and elastic modulus of deep-sea pressure resisting structure are required and they should be even higher by increasing the working depth [5,6,7]. However, the elastic modulus of titanium alloys is low, half of that of steel [8], limiting their use. Therefore, high elastic modulus titanium alloys are widely required. Ti-6Al-2Nb-2Zr-0.4B alloy is a new type of high elasticity modulus titanium alloy, developed according to the design theory of aluminum and molybdenum equivalents, and molecular orbital calculation of the electronic structure and β-phase stability coefficient. This alloy exhibited high elastic modulus and strength, good plasticity and toughness, and excellent comprehensive properties. The elastic modulus of the Ti-6Al-2Nb-2Zr-0.4B alloy can be considerably increased by the addition of a B element. However, the microstructure and properties of the alloy in high-temperature forging are sensitive to the deformation rate and temperature, and the hot-working window is relatively narrow. The material properties are closely related to its microstructure, which depends mainly on the processing method [9,10,11,12]. At present, the thermal deformation behavior of titanium alloys containing a B element has not been reported. Therefore, the evaluation of the deformation behavior of Ti-6Al-2Nb-2Zr-0.4B alloy has an important theoretical guiding significance and engineering application value for the processing of its forgings and plates and precise microstructure control.

In this study, the deformation behavior of Ti-6Al-2Nb-2Zr-0.4B alloy was studied by hot compressive test. The influences of strain rate and heating temperature on flow stress were also evaluated. The Arrhenius type thermal deformation constitutive equation was accomplished through the experimental data. The energy dissipation and plastic instability diagrams of the alloy were obtained, and the thermal working diagram was obtained by superposing the two diagrams. The deformation law of the alloy at high temperatures was analyzed and the suitable processing technology was obtained, providing a theoretical basis for the hot processing technology formulation.

## 2. Materials and Methods

The Ti-6Al-2Nb-2Zr-0.4B alloy studied in this paper was taken from a 60 mm diameter bar, which was hot-forged in the α + β range, and then a stress relief annealing was performed. The chemical composition indicated in Table 1 was calculated by JMatPro Software [13], and the composition was optimized by experiments. There are TiB phase and equiaxed primary α phase in a transformed β matrix in the optical microstructure, as shown in the Figure 1. The transit temperature (T_β_) was approximately 1002 °C. Additionally, ultimate tensile strength Rm and the yield strength Rp_0.2_ were 838 MPa and 745 MPa, respectively. 

The compression specimens with a diameter of 8 mm and a height of 12 mm were prepared by using Ti-6Al-2Nb-2Zr-0.4B alloy bars, as shown in Figure 2, and the concentric grooves with 7 mm in diameter and 0.2 mm in depth were fabricated on the bottom and top of the sample, which were used to retain the lubricant (10% graphite + 20% oil + 70% glass powder) in the deformation process, and to minimize friction and obtain the experimental conditions under the unidirectional stress state.

The compression tests were carried out at different temperatures (880, 910, 940, 970, 1000, 1030, and 1060 °C) and four different strain rates (0.01, 0.1, 1.0, and 10.0 s^−1^) using the Gleeble-3800 thermal simulator, and the compression ratio was 60%. The specimen was heated by electric frequency induction at a rate of 3 °C/s. The deformation temperature was held at the given temperature for 3 min.

During the compression process, a thermocouple welded in the middle of the sample was used to measure the temperature in real time. The temperature control accuracy is ±1.0 °C. The experimental data was collected and calculated using a testing machine equipped with an automatic microcomputer processing system, and finally outputs containing the load-stroke and flow stress–strain data were obtained in the form of a table. The specimen morphologies before and after deformation are shown in Figure 3 (conditions: deformation temperature, 970 °C; deformation rate, 1 s^−1^; deformation amount, 60%). Bulging was observed in the specimen after deformation; thus, the specimen is close to a unidirectional stress state.

## 3. Results and Discussion

### 3.1. Thermal Deformation Behavior

Assuming that the volume of the Ti-6Al-2Nb-2Zr-0.4B alloy remains constant during compression deformation, the load and displacement data were obtained using a computer. The true stress–strain curve of specimens during compression deformation can be obtained through Equations (1) and (2).
(1)$$\varepsilon_{tr}=\ln\left[L/(L_{0}–L\right)]\;$$
(2)$$\sigma_{tr}=F(L_{0}–L)/S_{0}L_{0}$$
where *L* is the displacement, *L*_0_ is the sample initial length, *F* is the load, and *S*_0_ is the sample sectional area.

Figure 4 shows the true stress–strain curves of Ti-6Al-2Nb-2Zr-0.4B alloy under different deformation conditions. The true stress–strain curves changed similarly and were relatively flat, indicating that all alloys presented satisfactory high-temperature deformation ability. At the same temperature, the flow stress increased with an increase in strain rate, indicating that the higher the deformation rate, the greater the deformation resistance. In the initial stage of deformation, due to the work hardening effect, the flow stress increased sharply due to the increase in strain, and it reached a peak at very low strain. In the process of thermal deformation, the dynamic recovery and dynamic recrystallization occurred. In the meantime, the softening mechanism was dominant, and the flow stress gradually decreased by increasing the strain. In addition, the content of *β* phase with 12 slip systems increased, the content of α phase with 3 slip systems decreased with an increase in temperature, and the slip system increased. The greater the possibility of metal slip, the lower the flow stress. Additionally, the steady–state flow characteristics of the alloy appeared under all deformation conditions during compression deformation, that is, when the true compression strain exceeded a constant value under certain strain rate conditions and deformation temperatures, the change of flow stress decreased and tended to a stable value by increasing the strain variable. At this point, the hardening and softening mechanisms reached dynamic equilibrium.

Figure 5 shows the true stress–strain curves of Ti-6Al-2Nb-2Zr-0.4B alloy at the strain rate of 1 s^−1^ and various deformation temperatures. At the same degree of deformation, the flow stress was strongly dependent on the temperature and it decreased by increasing the deformation temperature, but it was not a simple linear relationship. For example, the deformation resistance increased sharply under 940 °C. This is mainly because the atomic thermal vibration amplitude increased by increasing the temperature, the binding force between the atoms decreased, and the critical shear stress that causes slippage in the crystal decreased. Thus, the obstacles to the dislocation movement of materials and the sliding between crystalline surfaces were reduced. Simultaneously, the softening mechanisms easily occur by increasing the temperature, such as dynamic recovery and dynamic recrystallization, which reduces the dislocation density and counteracts the work hardening effect, reducing the flow stress of the alloy [14].

### 3.2. Constructive Equation

To accurately reflect the relationship between strain rate, deformation temperature, and flow stress in the process of thermal deformation, the Arrhenius type constitutive equation of Ti-6Al-2Nb-2Zr-0.4B alloy was determined in this paper. The equation can be expressed in three different manners [15].

At low-stress levels:(3)$$\dot{\varepsilon \;}=A_{1}\sigma^{n_{1}}\exp(-Q/RT)$$

At high-stress levels:(4)$$\dot{\varepsilon }=A_{2}{\exp(n}_{2}\sigma )\exp(-Q/RT)$$

Using hyperbolic sine functions for all stress levels:(5)$$\dot{\varepsilon }=A_{3}{[\sin h(\alpha \sigma )]}^{n}\exp(-Q/RT)$$
where *T* is the absolute temperature (K); *Q* is the deformation activation energy (J/mol); *R* is the universal gas constant (8.314 J/mol·K); *A*_1_, *A*_2_, and *A* are the material constants; n and n_1_ are the stress exponents. Moreover, α, $n_{1}$, and $n_{2}$ satisfy α = $n_{2}$/$n_{1}$ [16,17,18].

When *T* is a constant, it can be obtained from Equations (3) and (4) as follow: (6)$$n_{1}=\partial \ln\dot{\varepsilon }/\partial {\ln\sigma }_{p}$$
(7)$$n_{2}=\partial \ln\dot{\varepsilon }/\partial \sigma_{p}$$

The relation diagrams of $\partial {\ln\sigma }_{p}–\partial \ln\dot{\varepsilon }$ and $\partial \sigma_{p}–\partial \ln\dot{\varepsilon }$ were obtained using strain rate and the strain peak, as shown in Figure 6. The average slopes of the lines were calculated in Figure 6, the n_1_ value was 6.95, the $n_{2}$ value was 0.072, the α value was 0.01.

The material needs to overcome a specific potential barrier to complete the thermal deformation process. The potential barrier is the activation energy during the hot deformation, and its magnitude reflects the difficulty of dislocation initiation, recovery, and recrystallization during the hot deformation of the alloy.

From Equation (5), the activation energy *Q* for deformation can be obtained as:(8)$$Q=R\left\{\frac{\partial \ln\dot{\varepsilon }}{\partial \ln[{\sin h(\alpha \sigma }_{p})]}\right\}\left\{\frac{\partial \ln[{\sin h(\alpha \sigma }_{p})]}{\partial \left(1/T\right)}\right\}$$

The $\ln\dot{\varepsilon –}\ln[{\sin h(\alpha \sigma }_{p})]$ curve under different temperature conditions and the $1/T-\ln[{\sin h(\alpha \sigma }_{p})]$ curve under different strain conditions of Ti-6Al-2Nb-2Zr-0.4B alloy were plotted according to the compressive test, as illustrated in Figure 7. The relationships between $\ln[{\sin h(\alpha \sigma }_{p})]$ or $\ln\dot{\varepsilon }$ and 1/*T* were approximately linear, indicating that the hyperbolic sinusoidal Arrhenius equation satisfactorily describes the thermal deformation behavior of Ti-6Al-2Nb-2Zr-0.4B alloy. The slope of the fitting line in Figure 7 is calculated, and then substituted in Equation (8) to obtain the hot deformation activation energy *Q* of 551,706 J/mol (the standard deviation is 45.2 J/mol), which is much larger than the activation energy of pure *α* titanium (204 kJ/mol) and pure β titanium (161 kJ/mol) [19]. The activation energy of pure α titanium is higher than that of pure β titanium, this is because the activation energy is related to the microstructure of the material. Pure α titanium is HCP structure with three slip systems, and pure β titanium is BCC structure with 12 slip systems. With more slip systems, the greater the possibility of metal slipping, the more prone to plastic deformation, and the lower the activation energy required. For Ti-6Al-2Nb-2Zr-0.4B alloy, the activation energy increases with an increase in alloying elements content in titanium due to the solid solution strengthening of alloying elements. More importantly, tiny TiB (as shown in Figure 1) particles are generated due to the addition of element B, which have a pinning effect on dislocation. This greatly increases the activation energy of the alloy. From a macroscopic point of view, the elastic modulus scale indicates the difficulty of the material elastic deformation under certain stress. The larger the elastic modulus, the more difficult the deformation [20,21,22]. From a perspective of the interaction force between atoms, the elastic modulus represents the strength of the interatomic bonding force. The greater the elastic modulus, the greater the binding force between atoms. By adding 0.4B element, the elastic modulus of Ti-6Al-2Nb-2Zr-0.4B alloy greatly increased, which is essentially the interatomic bonding force increase. Therefore, the thermal deformation activation energy greatly increased. This indicates that the Ti-6Al-2Nb-2Zr-0.4B alloy can initiate less slip system at a low temperature, and the dislocation can produce packing at the grain boundary and other defects, which cannot be effectively released by the recovery mechanism controlled by diffusion. Under these conditions, the thermal deformation of the alloy is controlled by a process other than high-temperature diffusion, and the dynamic recovery and recrystallization are identified [23,24].

Zener and Hollomon proposed a “temperature compensated strain rate” to verify the relationship between strain rate and deformation temperature *T* in high-temperature experiments, which can be expressed using the parameter *Z* [25,26]:(9)$$Z\;=\dot{\varepsilon }\exp(Q/RT)=A_{3}{[\sin h(\alpha \sigma )]}^{n}$$

From Equation (9), the ln *Z* can be obtained as:(10)$$\ln\;Z=n\ln[\sin h(\alpha \sigma )]\;+\;\ln A_{3}$$

The relation graph of ln[sinh(ασ)]  and ln *Z* was determined, as illustrated in Figure 8, and linear regression was performed on the data. The obtained *A*_3_ was e^54.41^ s^−1^ and *n* equalled 2.35. The uncertainties of *n* and ln *A*_3_ can be seen in the table in Figure 8. By combining with Equation (5), the hyperbolic sinusoidal Arrhenius equation of the Ti-6Al-2Nb-2Zr-0.4B alloy at a deformation temperature range of 850–1000 °C and strain rate range of 0.01–10.0 s^−1^ was obtained:(11)$$\dot{\varepsilon \;}{=e}^{54.41}{[\sin h\;(0.01\sigma )]}^{2.35}\exp(-551.7/RT)\;$$

### 3.3. Processing Maps

#### 3.3.1. Dissipative Judgment Criterion

According to the dynamic material model, the fragment subjected to deformation is a nonlinear energy dissipation unit. The energy input by external forces to the fragments contributes mainly two aspects: (1) plastic deformation, most of this is converted to heat and a small amount is stored in materials and (2) microstructure evolution, as recovery, recrystallization, and phase transition. The energy dissipated by microstructure evolution is denoted by *J* and the energy dissipated by plastic deformation is represented by *G*. Among them, the process of plastic instability and fracture is related to *G*, while the microstructure evolution is related to *J* [18]. The energy *P* absorbed by the fragments in the process of plastic flow can be expressed as [27,28]:(12)$$P\;=\bar{\sigma }\dot{\bar{\varepsilon }}=\int_{0}^{\bar{\sigma }}\dot{\bar{\varepsilon }}d\bar{\sigma }+\int_{0}^{\dot{\bar{\varepsilon }}}\bar{\sigma }d\dot{\bar{\varepsilon }}=J+G$$
where $\;\dot{\bar{\varepsilon }}$ is equivalent strain rate (s^−1^) and $\bar{\sigma }$ is equivalent flow stress (MPa). The portioning of power between *J* and *G* is given by [29,30]:(13)$$m=\frac{dJ}{dG}=\frac{\dot{\bar{\varepsilon }}d\bar{\sigma }}{\bar{\sigma }d\dot{\bar{\varepsilon }}}=\frac{d\log\bar{\sigma }}{d\log\dot{\bar{\varepsilon }}}$$
where m is the strain rate sensitivity parameter. The parameter m can be regarded as the energy distribution index, which consumes the total absorbed energy in a specific proportion to the microstructure evolution *J* and plastic deformation G, respectively.

Under certain deformation strain and temperature, *J* can be given by the following equation [31]:(14)$$J=\frac{m}{m+1}\cdot \sigma \dot{\cdot \varepsilon }$$
where $\dot{\varepsilon }$ is strain rate (s^−1^) and *σ* is flow stress (MPa).

For an ideal linear dissipative unit, $\;J_{\max}=\sigma \cdot \dot{\varepsilon }/2$ and *m* = 1. The nonlinear dissipation element and ideal linear dissipation element are normalized to produce a dimensionless parameter, which is expressed as [32]:(15)$$\eta \;=\frac{J}{J_{\max}}=\frac{2m}{m+1}$$

The parameter *η* reflects the mechanism of microstructure change within a specific range of strain rate and temperature. The variation of *η* with temperature and strain rate generates the power dissipation diagram of different regions. For material processing, dynamic recrystallization, dynamic recovery, and superplasticity are safe thermal deformation mechanisms, while the formation of wedge-shaped cracks and voids is dangerous and should be avoided. Therefore, the power dissipation rate *η* plays a very important role in selecting the best processing parameters in hot processing.

#### 3.3.2. Plastic Instability Criterion

The criterion of material rheological instability was proposed by Kumar based on Zeigler’s principle of maximum entropy production rate [33]:(16)$$\xi \left(\dot{\varepsilon }\right)=\frac{\partial \log\frac{m}{m+1}}{\partial \log\;\dot{\varepsilon }}+m\;<\;0$$
where $\;\xi \left(\dot{\varepsilon }\right)$ are the stability functions.

When Equation (16) is satisfied, an unstable rheological process occurs. The parameter $\xi \left(\dot{\varepsilon }\right)$ is taken as the function of strain rate and deformation temperature, and the corresponding $\xi \left(\dot{\varepsilon }\right)$ values of each strain rate and deformation temperature, are calculated to form the instability diagram.

#### 3.3.3. Processing Maps Analysis

According to the above principles and methods, the *m* parameter, the *η* rate due to the microstructure change, and the $\xi \left(\dot{\varepsilon }\right)$ parameter for each variable strain rate and temperature are calculated, respectively. The instability maps and power dissipation are generated through the parameters *ξ* and *η*, which are used as functions of the strain rate and deformation temperature, respectively. The final thermal processing maps are generated by the superposition of the instability maps and power dissipation maps [34].

When the true strain ε is 0.5, the relationship between logσ  and $\log\;\dot{\varepsilon }$ of the alloy at four strain rates and eight deformation temperatures is presented in Figure 9.

The curve between log σ and $\log\dot{\varepsilon }$ in Figure 9 is fitted using the cubic spline function. The magnitude of logσ can be indicated by the polynomial in $\log\dot{\varepsilon }$ [35]:(17)$$\log\sigma =a+\mathrm{blog}\dot{\varepsilon }+c{\left(\log\dot{\varepsilon }\right)}^{2}+d{\left(\log\dot{\varepsilon }\right)}^{3}$$
(18)$$m=\frac{d(\log\sigma )\;}{d(\log\dot{\varepsilon })}=b+2\mathrm{clog}\dot{\varepsilon }+3d{\left(\log\dot{\varepsilon }\right)}^{2}$$

The values of *m* at the corresponding points can be determined by substituting the values of $\dot{\varepsilon }$ at each point of Equation (18). Thus, the *η* rate of the alloy at different strain rates and deformation temperatures is get by substituting the values of *m* in Equation (15). The power dissipation diagram under the whole experimental condition can be obtained by the interpolation method, as shown in Figure 10.

The values of $\xi \left(\dot{\varepsilon }\right)$ are obtained by substituting the values of $\log\dot{\varepsilon }$ and *m* for various conditions in Equation (19) and the alloy instability map can be obtained using the interpolation method [36], as shown in Figure 11.
(19)$$\xi \left(\dot{\varepsilon }\right)=\frac{\partial \log\frac{m}{m+1}}{\partial \log\dot{\varepsilon }}+m=\frac{2c+6d(\log\dot{\varepsilon })}{m(m+1)\ln10}+m$$

The processing maps of the alloy at ε = 0.5 can be obtained by superposing the contour maps of Figure 10 and Figure 11, as illustrated in Figure 12. The contour value is the *η* rate value and the shaded parts represent the instability areas in Figure 12. If the alloy is plastically deformed under the process parameters corresponding to the instability zone, several defects that are detrimental to the microstructure may occur. Thus, hot working in this zone should be avoided. Figure 12 shows two regions in the safe zone, in which the *η* rate value is larger. In (α + β) two-phase region, the power dissipation rate reaches 53% in the range of 0.01–0.05 s^−1^ and temperature range of 920–980 °C, and the deformation mechanism is dynamic recovery. In β single-phase region, the power dissipation rate reaches 48% in the range of 0.01–0.1 s^−1^ and temperature range of 1010–1040 °C, and the deformation mechanism is also dynamic recovery. The results show that these two regions are the best deformation parameters of Ti-6Al-2Nb-2Zr-0.4B alloy under laboratory conditions. The instability zone is mainly in the temperature range 890–940 °C and strain rate range 0.1–10.0 s^−1^. When the deformation temperature is below 940 °C, the deformation resistance is high and it is not suitable for low-temperature forging. When the strain rate is equal or greater than 0.1 s^−1^, it is easy to crack due to the B element influence and it is not suitable for high strain rate forging, which is consistent with the forging experimental results.

## 4. Conclusions

The true stress–strain curves were similar under all conditions. The flow stress reached the peak fast (hardening) by increasing the strain followed by a gradual decrease (softening), exhibiting a typical dynamic recrystallization curve. When the true compressive strain exceeded a certain value, the change in flow stress decreased and tended to a stable value by increasing the strain variable.The thermal deformation activation energy Q was 551.7 kJ/mol, and the constitutive equation was $\dot{\varepsilon \;}{=e}^{54.41}{[\sin h\;(0.01\sigma )]}^{2.35}\exp(-551.7/RT)\;$ hyperbolic sine functions.The suitable hot-working zone of the alloy was in the range of 0.01–0.05 s^−1^ and temperature range of 920–980 °C at the (α + β) two-phase region, and the energy dissipation rate was 53%. The other zone was in the range of 0.01–0.1 s^−1^ and temperature range of 1010–1040 °C at the β single-phase region, and the energy dissipation rate was 48%.

## Figures and Tables

**Figure 1 materials-14-02456-f001:**
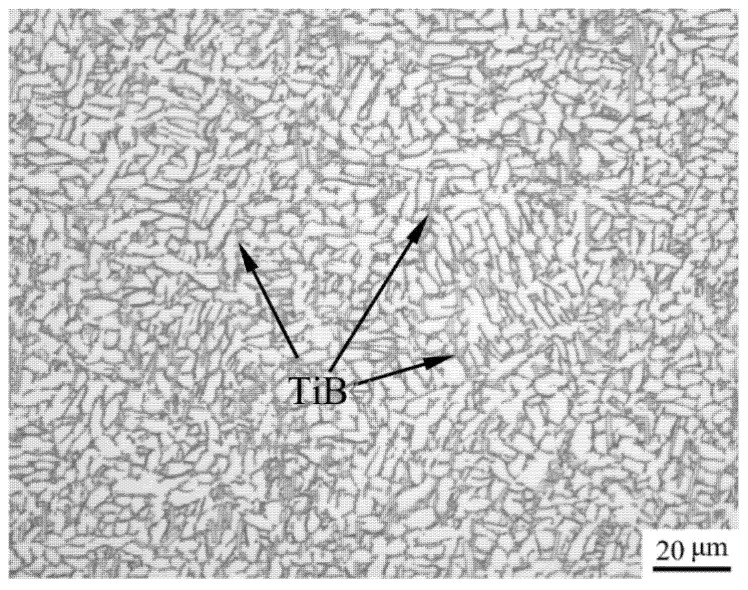
Optical microstructure of the Ti-6Al-2Nb-2Zr-0.4B titanium alloy in the as-received condition.

**Figure 2 materials-14-02456-f002:**
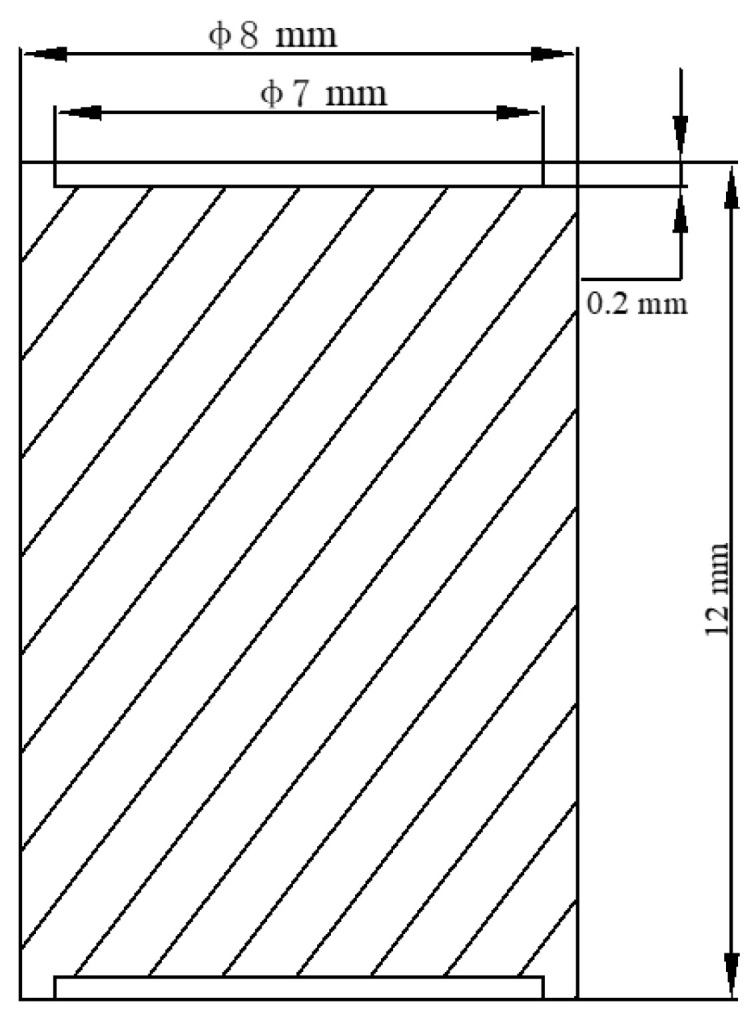
Schematic diagram of the compression sample.

**Figure 3 materials-14-02456-f003:**
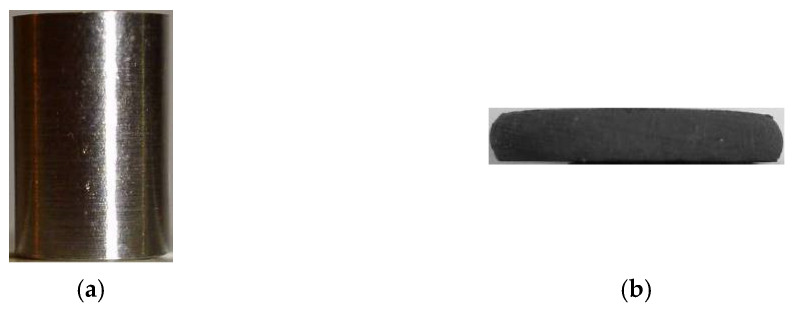
Compression specimen morphology: (**a**) Before deformation; (**b**) after deformation.

**Figure 4 materials-14-02456-f004:**
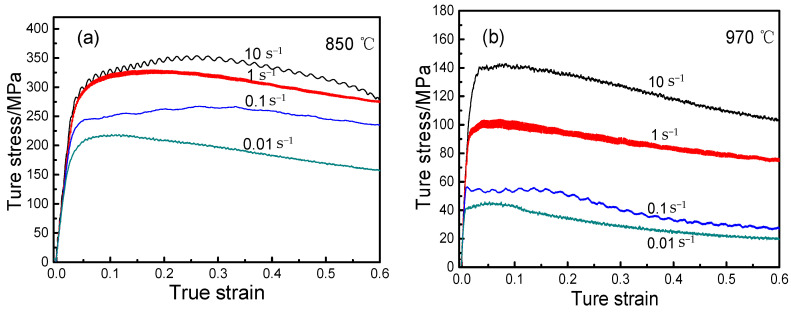
As-measured true stress–strain curves of the Ti-6Al-2Nb-2Zr-0.4B alloy: (**a**) At 850 °C for different strain rates; (**b**) at 970 °C for different strain rates.

**Figure 5 materials-14-02456-f005:**
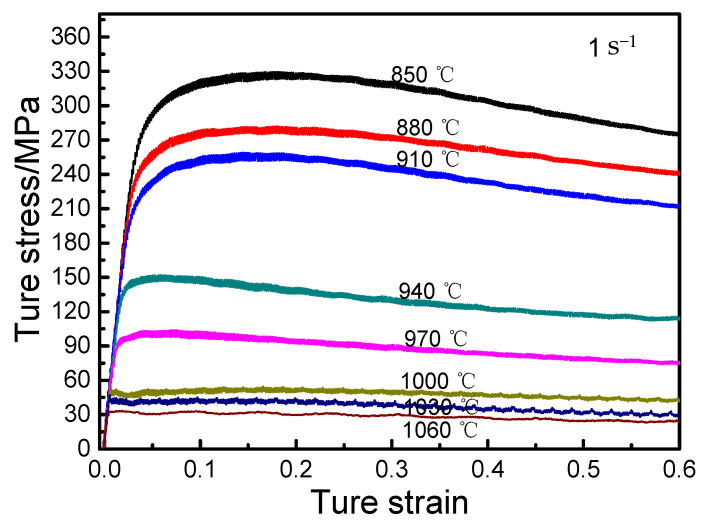
True stress–strain curves at different temperatures ($\dot{\varepsilon }$ = 0.1 s^−1^).

**Figure 6 materials-14-02456-f006:**
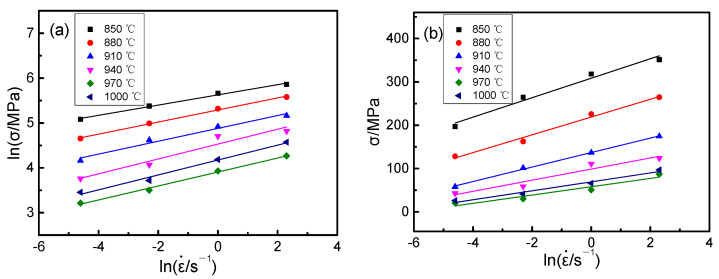
Relationships between $\ln\dot{\varepsilon }$ and ${\ln\sigma }_{p}$ or $\sigma_{p}$: (**a**)$\;\partial {\ln\sigma }_{p}–\partial \ln\dot{\varepsilon }$; (**b**) $\partial \sigma_{p}–\partial \ln\dot{\varepsilon }$.

**Figure 7 materials-14-02456-f007:**
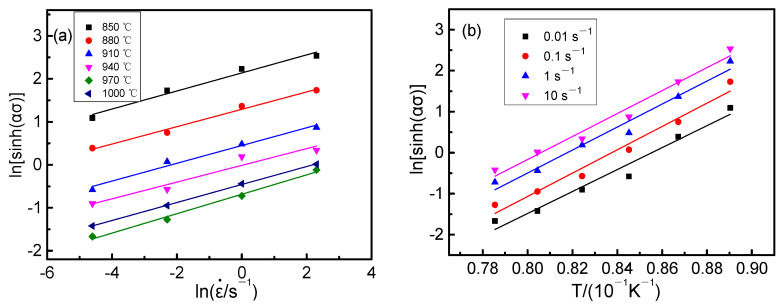
Relationships between $\ln[{\sin h(\alpha \sigma }_{p})]$ and $\ln\dot{\varepsilon }$ or 1/*T*: (**a**)$\;\ln[{\sin h(\alpha \sigma }_{p})]–\ln\dot{\varepsilon }$; (**b**) $\ln[{\sin h(\alpha \sigma }_{p})]–1/T$.

**Figure 8 materials-14-02456-f008:**
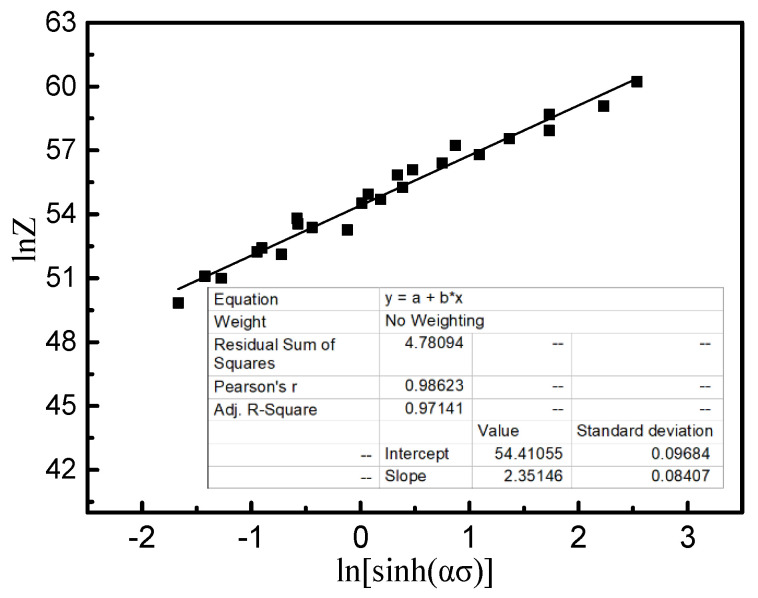
Relationships between ln[sinh(ασ)] and lnZ.

**Figure 9 materials-14-02456-f009:**
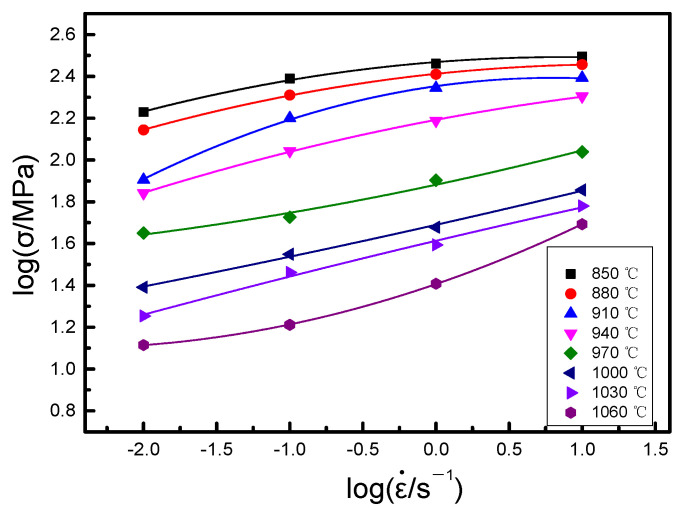
The $\log\sigma -\log\dot{\varepsilon }$ curves at different temperatures.

**Figure 10 materials-14-02456-f010:**
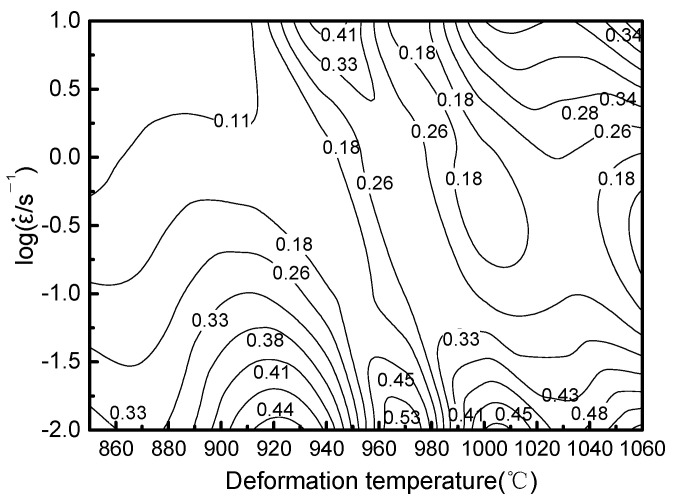
Power dissipation map of the Ti-6Al-2Nb-2Zr-0.4B alloy.

**Figure 11 materials-14-02456-f011:**
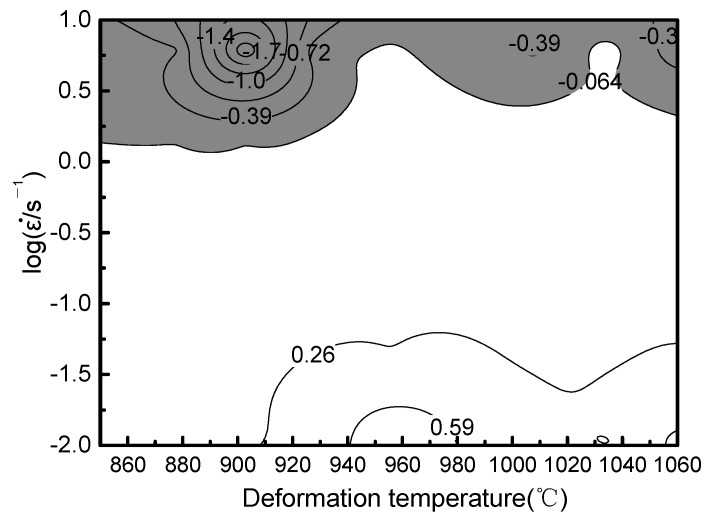
Instability map of the Ti-6Al-2Nb-2Zr-0.4B alloy.

**Figure 12 materials-14-02456-f012:**
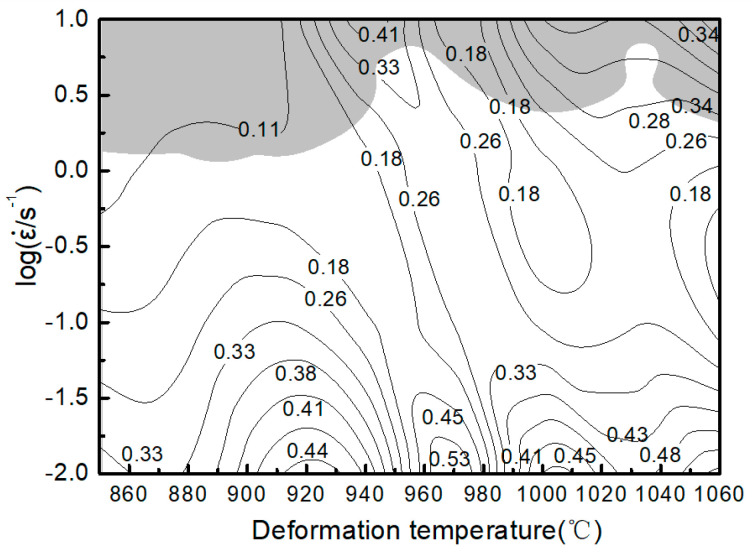
Processing maps of the Ti-6Al-2Nb-2Zr-0.4B alloy.

**Table 1 materials-14-02456-t001:** Chemical composition of the Ti-6Al-2Nb-2Zr-0.4B alloy (wt.%).

Al	Nb	Zr	B	Si	C	N	H	O
5.97	2.01	1.93	0.402	<0.010	0.029	0.0062	0.0010	0.074

## Data Availability

The data presented in this study are available on request from the corresponding author.

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
