# Peer review of "Hot Deformation Behavior and Processing Maps of a New Ti-6Al-2Nb-2Zr-0.4B Titanium Alloy"

_materials, 2021, doi:10.3390/ma14092456_

Round 1

Reviewer 1 Report

This paper provides the adequate data set of the flow stress of the advanced Ti alloy. As mentioned in the introduction of this paper, the advanced Ti alloy has been studied aggressively and the related data should gather much attention from many researchers. So that this paper is worth for the publication. There are some unclear points some of which should be revised.

(1) The presentation of the significant figure should be revised. The 5 digits or more seem not to be reasonable.

(2) (P.1, L.7 at the 1st paragraph of the introduction) elasticity

'Ti-6Al-2Nb-2Zr-0.4B alloy is a new type of high elastic modulus titanium alloy, developed according to...'

The reference works should be necessary because the elasticity of the studied alloy is mentioned not only at the introduction but also at the discussion of the results (P.6, L.17, 'The larger elastic modulus, ...')

(3) (Experiments, P.2, last paragraph)

What kind of lubricant did this work use?

(4) (Experiments, P.3, last sentence. 'No obvious drum...'

'Bulging' should be more suitable to present the shape of the pressed cylinder.

(5) (Fig.4 and P.4, L.8. 'The dynamic recovery and ...')

The deformation is conducted at the multi-phase region so that the reconstructive transformation possibly occurs. Such microstructure change influences the flow stress as well, although this paper did not discuss this possibility.

(6) (P.5, L.29. '... were obtained using the strain peak...')

'The strain pear' should be (the stress peak, sigma_p'

(7) (the deformation mechanism, P.6, L.3)

'... and its magnitude reflects the difficulty of dislocation initiation, recovery...'

The possible thermal activation processes with the dislocation slip is not limited at the initiation but also the jumping process overcoming different kinds of obstacles. Some of these obstacles is related to the microstructure. However, this paper does not mention this relation between the activation energy and the microstructure.

(8)Power dispersion map

The processing maps have been reported for some Ti alloys by several papers such as Y.V.R.K. Prasad, T. Seshacharyulu, Mater. Sci. Eng. A, 243(1998) 82-88. The map reported by this paper looks very different from those which reported previously. These difference should be discussed.

Reviewer 2 Report

The manuscript reports an experimental study of Ti-6Al-2Nb-2Zr-0.4B high elastic modulus titanium alloy, focused on its hot deformation behaviour. The key result is the construction of the constitutive equation for the alloy based on the parameters determined from the measurements. Moreover, the processing maps are constructed.

The paper is interesting and convincing, the results are of importance to the community.

I would recommend the manuscript for publication in Materials journal, provided that the Authors give prior consideration to the points raised below (mainly concerning some aspects of data presentation):

Page 2: it might be useful for the Reader to mention how the chemical composition indicated in table 1 was determined.

Page 3, eq. (1): please correct carefully the form of the equation – is logarithm (ln) function applied to the right hand-side? If yes, it looks misleading, as it merges with the next symbol.

Figures 4 and 5: please correct “ture” -> “true” in some places.

Page 5, eq. (5) and further discussion: please use a single symbol for a / alpha – the different letters used in the equation and the further discussion (and further equations) may look misleading.

Page 5: because sigma is expressed in units of pressure (typically MPa), the quantities n2 and alpha are not dimensionless (contrary to n1). Therefore, the values determined from the fitting should have appropriate units. Moreover, it should be possible to give also the uncertainties of the fitting parameters (standard deviations).

Page 6: can the determined value of Q be also accompanied with the uncertainty?

Figures 6, 7, 9, 10, 11 and 12: I believe that in the description of the horizontal axis, where ln epsilon dot is present, it should be mentioned that epsilon dot is in 1/s units.

Figure 8: the unit of Z is 1/s, so that it should be mentioned in the description of the vertical axis where ln Z is present. Moreover, the product alpha*sigma seems dimensionless (as the dimension of alpha is 1/dimension of stress). Therefore, I guess it is not necessary to indicate that sigma is in MPa here.

Page 7, Figure 8 and its discussion: the values of n and ln A3 determined from fitting might also be accompanied with the uncertainties.

Page 7, eq. (9): the parameter Z might be called a Zener-Hollomon parameter and the original work might be cited (doi: 10.1063/1.1707363).

Page 7: I guess the unit of A3 is 1/s, so it should be mentioned when the value is given.

Figures 10, 11 and 12: the description of vertical axis should be corrected (temperature in K).

Page 7: After eq. (11), it should be stated that it is valid for T in K, sigma in MPa and epsilon dot in 1/s. Alternatively, the division of the quantity by the unit can be incorporated directly into the equation.

Abstract: the unit of Q should be written as kJ/mol (not KJ/mol).

Reviewer 3 Report

The work which I reviewed addresses thermal deformation investigation of an titanium alloy which contain boron element. Such research is not reported up to date. The boron addition is believed to increase elastic modulus of titanium alloy. The importance of titanium based alloys is obvious therefore the manuscript address important topic and can be considered for publication after addressing below given remarks.

There could be more extensive literature research. For now there is only 24 references.

Table 1 “(mass fraction, %).” Mass fraction is by definition is in the range 0 to 1. I advise to use wt.%.

3.2 constructive equation:

“where Q is the deformation activation energy (kJ/mol), T is the absolute temperature (K), R is the universal gas constant (8.314 J/mol K), “ – please note that unit dimension should be the same for Q and R in order to obtain dimensionless value. This can be a reason of very high i.e. -1111 kJ/mol activation energy Q. I am a chemist and the strongest chemical bonding (nitrogen molecule) is 940 kJ/mol. Therefore I am sceptical about the value presented in the manuscript. Please look on the results of other researchers whether this value is not at least one order of magnitude too high. Another problem is uncertainty of measurement. Your result 1111.883 kJ/mol in my opinion is far to precise. I guess the measurement were delivering results justifying the precision of 1112 kJ/mol. In order to determine this problem standard deviation of measurement should be evaluated.

The table in Fig. 8 is hardly readable (to small)

Round 2

Reviewer 1 Report

n/a

Reviewer 3 Report

The Authors improved substantialy the manuscript acordingly to my suggestions. Therefore I recomend it for publication.